# Polyphenols in Jabuticaba (*Plinia* spp.) Peel Flours: Extraction and Comparative Evaluation of FTIR and HPLC for Quantification of Individual Compounds

**DOI:** 10.3390/foods12071488

**Published:** 2023-04-01

**Authors:** Laís M. Resende, Leandro S. Oliveira, Adriana S. Franca

**Affiliations:** 1PPGCA, Food Science Graduate Program, Universidade Federal de Minas Gerais, Av. Antônio Carlos, 6627, Belo Horizonte 31270-901, MG, Brazil; lais.maia.resende@gmail.com (L.M.R.);; 2DEMEC, Universidade Federal de Minas Gerais, Av. Antônio Carlos, 6627, Belo Horizonte 31270-901, MG, Brazil

**Keywords:** jabuticaba, cyanidin-3-*O*-glucoside, ellagic acid, delphinidin-3-*O*-glucoside, FTIR, PLS

## Abstract

Jabuticabas are wild fruits native to Brazil, and their peels, the main residue from jabuticaba processing, contain significant amounts of bioactive compounds, which are mostly phenolics. Conventional methods based on the estimation of total extractable phenolics (TEP—Folin–Ciocalteau) or total monomeric anthocyanins (TMA) have limitations and may not reflect the actual antioxidant potential of these peels. Analytical methods, such as high-performance liquid chromatography (HPLC), are more appropriate for the quantification of specific phenolics, and can be used as a reference for the construction of mathematical models in order to predict the amount of compounds using simple spectroscopic analysis, such as Fourier Transform Infrared Spectroscopy (FTIR). Therefore, the objectives of this study were (i) to evaluate the composition of specific polyphenols in flours prepared from jabuticaba peels and verify their correlation with TEP and TMA results from a previous study, and (ii) to employ FTIR coupled with chemometrics to predict the concentrations of these polyphenols in jabuticaba peel flours (JPFs) using HPLC as a reference method. Cyanidin-3-glucoside (C3G), ellagic acid (EA) and delphinidin-3-glucoside (D3G) were the main polyphenols found in the samples. The C3G contents ranged from 352.33 mg/100 g (S10) to 1008.73 mg/100 g (S22), with a strong correlation to TMA (r = 0.97; *p* = 0.00) and a moderate correlation to TEP (r = 0.45; *p* = 0.02). EA contents ranged from 163.65 mg/100 g (S23) to 334.69 mg/100 g (S11), with a moderate to strong correlation to TEP (r = 0.69; *p* = 0.00). The D3G values ranged from 94.99 mg/100 g (S10) to 203.36 mg/100 g (S5), with strong correlations to TMA (r = 0.91; *p* = 0.00) and C3G levels (r = 0.92; *p* = 0.00). The developed partial least squares-PLS models based on FTIR data provided satisfactory predictions of C3G and EA levels, reasonably matching those of HPLC.

## 1. Introduction

Polyphenols comprise a diverse class of compounds with antioxidant activity and several health benefits. They are secondary plant metabolites that are located mainly in the vacuoles of cells and act in the plant’s defense system against pathogen aggression, temperature changes, and ultraviolet radiation, among other threats. Some common sources of polyphenols include cocoa and its products, dried herbs, hazelnut, olive, artichoke, flaxseed and oilseeds, such as chestnuts and dark-colored berries [1,2].

Jabuticaba (*Plinia* sp.) (Figure 1), a Brazilian fruit, is an example of a dark-colored fruit that is a source of polyphenols, especially anthocyanins, which are responsible for the dark purple to black color of its peel. Anthocyanins and hydrolysable tannins were quantified in high amounts in both *Plinia jaboticaba* and *Plinia trunciflora* species by Quatrin et al. [3]. Regardless of the species, the phenolic compound found in greater quantity was cyanidin-3-*O*-glycoside (C3G). Neves et al. [4] also quantified high amounts of anthocyanins in *P. jaboticaba*, which were mainly C3G and delphinidin-3-*O*-glucoside (D3G), in addition to other compounds such as flavonols, derived from quercetin and myricetin, and a wide variety of ellagic acid and methyl ellagic acid derivatives.

However, it is known that the content of polyphenols can vary considerably in fruits of the same species due to environmental conditions and agricultural practices. In a previous study, it was shown that there was a significant variation in the amount of total polyphenols, monomeric anthocyanins, proanthocyanidins and carotenoids in jabuticaba peel flours prepared from samples harvested at different municipalities from the state of Minas Gerais, Brazil [5]. Furthermore, Fernandes et al. [6] highlighted that the variation in anthocyanin content between food sources, due to different environmental conditions, is neglected in most studies, evidencing the existing knowledge gap in this area.

Studies comparing samples of different species were found in the literature, but the investigation of specific polyphenols from a large sample of jabuticaba peels from the same region has not yet been described. It is expected that the polyphenol content should vary among jabuticaba peels of different species, including some rare ones. However, the variation between samples that come from distinct places within a given region has not yet been evaluated. It is important to notice that, for this type of comparison, all samples should be evaluated by the same method and the extraction of polyphenolics should be performed in the same way.

The quantification of specific polyphenols requires a more robust method of analysis. High Performance Liquid Chromatography (HPLC) has been widely used for the identification and quantification of polyphenols; however, it is a high-cost method, with high time demands, and generates toxic wastes. Therefore, Fourier-Transform Infrared (FTIR) spectroscopy, together with chemometric models, has been tested for the prediction of compounds for which the reference analysis method is HPLC. Deus et al. [7] used FTIR and PLS regression to quantify bioactive amines in dark chocolates and obtained models with good performance, especially for spermidine and tryptamine. Furthermoe, in chocolates, Hu et al. [8] obtained good catechin and epicatechin prediction models employing PLS and FTIR. The combination of these two methods was also shown to be a good alternative for the quantification of anthocyanins [9] and ellagitannins [10] in wines.

Many studies still employ UV-visible spectroscopy to predict total polyphenols or a class of them, such as anthocyanins [5,11,12,13]. Studies correlating such measurements with the prediction of specific compounds, quantified by chromatography, are still scarce and none have been presented so far for jabuticaba peels.

Therefore, the objectives of this work were to evaluate the composition of specific polyphenols in flours prepared from jabuticaba peels (JPFs), investigate how much the content of these compounds can vary in samples produced in the same region and employ FTIR together with chemometrics to predict the concentrations of these polyphenols in JPF. Although there are several studies on the chemical characterization of phenolics in jabuticaba peels, this is the first study that attempts to quantify phenolics by FTIR for this specific food byproduct. The possibility of a simpler and reliable evaluation of these compounds can contribute to routine analyses, reduce costs and limit waste generation and the time spent for analysis.

In summary, the major contributions of this study are as follows: (i) This is the first work that provides an investigation of specific polyphenols from a large sample of jabuticaba peels from distinct microregions of the same Brazilian state, and, thus, the first to report on the intrinsic variability of the samples attributed to differences in edaphoclimatic conditions and agricultural practices. (ii) This is the first work to correlate HPLC and FTIR measurements for the quantification of specific compounds in jabuticaba peel flours, including chemometric analysis and model building for a large sample of peels.

## 2. Materials and Methods

### 2.1. Materials

Twenty-eight jabuticaba samples were acquired from distinct microregions in Minas Gerais State, Brazil (see Appendix A). HPLC grade acetic acid, yanidin-3-*O*-glucoside (C3G) and ellagic acid (EA) were purchased from Sigma-Aldrich (São Paulo, Brazil). HPLC grade ethanol and methanol were purchased from Panreac AppliChem and Honeywell, respectively (São Paulo, Brazil). Formic acid was obtained from Êxodo Científica (Sumaré, Brazil).

### 2.2. Sample Preparation

The ripe fruits were selected (visual inspection), washed and sanitized. The peels were manually separated and dried in a convective oven at 60 °C for 20 h. Samples were then ground and sieved (425 μm) and stored at −18 °C, and will be herein denominated jabuticaba peel flours, JPFs [5]. It is noteworthy that drying was used as a strategy in terms of conservation and application in order to increase the shelf life and safety of the product as a food ingredient, and also to concentrate the content of bioactive compounds. Although high temperatures can affect the polyphenol content, the conditions employed in our study were based on the findings reported by Larrauri et al. [14] on the effect of drying temperature on the stability of polyphenols in red grape pomace, and are within the range that is commonly employed for producing flours based on fruit and vegetable byproducts [15].

### 2.3. Extraction of Polyphenols

Polyphenols were extracted following the procedure outlined by Barros et al. [16], with adaptations. Briefly, 0.1 g of each JPF sample was mixed with 5 mL of 50% ethanol (*v*/*v*) acidified with acetic acid (final pH 3.0 ± 0.1) in tubes protected from light. These were then sonicated for 60 min at 30 °C using an ultrasonic bath. The extracts were then centrifuged at 3500 rpm for 5 min and the remaining solids were mixed with 50% ethanol (*v*/*v*) and centrifuged for 5 min. This step was performed twice. The supernatants were combined and the pH adjusted to 3.0 with acetic acid. The final volume of the solution was 10 mL.

### 2.4. Chromatographic Analysis

The profile of phenolic compounds was investigated by HPLC according to the methods described by Plaza et al. [17], with modifications. The extracts were diluted with ultrapure water (final concentration of 0.0025 g/mL) and filtered using a 0.22 µm syringe filter. Analysis was performed using a chromatograph (Shimadzu, Japan) with DAD detector and C18 reversed phase column (5 µm particle size, 4.6 µm × 150 mm) at 30 °C. The injection volume was 20 μL, and the mobile phases consisted of 0.5% (*v*/*v*) aqueous formic acid (solvent A) and 0.5% (*v*/*v*) formic acid in methanol (solvent B). The elution gradient was as specified in Table 1, with subsequent cleaning and conditioning of the column, at a flow rate of 0.6 mL/min. 

The signals at 520 nm and 350 nm were registered using a UV-vis detector. The previous literature [18,19] has reported C3G, EA and D3G as the main phenolics found in jabuticaba peels, so these compounds were identified by comparing the retention times and spectra of authentic standards (C3G and EA) and, in the case of delphinidin-3-*O*-glucoside (D3G), the spectrum reported in the literature. The standards were used to construct calibration curves for the quantification of compounds, as specified in the following equations:
C3G:          y = 85417x − 149480 (R² = 0.9974)(1)
EA:          y = 24526x − 16202 (R² = 0.9996)(2)

D3G was quantified by the C3G standard curve, as performed by Plaza et al. [17]. Concentrations were expressed as mg polyphenol/100 g JPF. 

### 2.5. FTIR Analysis

JPF samples with particle size < 0.15 mm were analyzed in a Shimadzu IRAffinity-1 FTIR Spectrophotometer (Shimadzu Corporation, Kyoto, Japan), using an ATR sampling accessory (MIRacle), as described by Resende, Oliveira and Franca [5]. Spectra were recorded in the 4000–600 cm^−1^ wavenumber range with 4 cm^−1^ resolution and 20 scans in dry atmosphere (20 ± 0.5 °C). They were submitted to background subtraction (atmosphere spectra) and atmospheric correction (carbon dioxide and water) using the IRsolution Ver.1.20 software (Shimadzu Corporation Kyoto, Japan).

### 2.6. Statistical Analysis

All samples were analyzed in triplicate and results were expressed as mean ± standard deviation. Normality was verified by the Shapiro–Wilk method. Significant differences were investigated by ANOVA and Tukey tests, and by the nonparametric Kruskal–Wallis test for D3G, with 95% confidence (*p* < 0.05), using IBM SPSS Statistics software, version 19.

Pearson correlations were evaluated between the values obtained for polyphenols and the parameters of total extractable phenolics, total monomeric anthocyanins and non-extractable polyphenols, and the parameters of color luminosity, chroma and hue angle, available from a previously published paper that employed the same set of samples [5].

Principal component analysis (PCA) was performed to evaluate similarities and differences between JPFs in terms of polyphenol profile and content. PLS regression models were constructed to predict polyphenol contents measured by HPLC. For these analyses, MATLAB software, version R2009b (MathWorks, Natick, MA, USA) and PLS Toolbox version 4.0 (Eigenvector Technologies, Wenatchee, WA, USA) were used. The data used for PCA were autoscaled. For PLS, spectral data were subjected to the following preprocessing techniques: baseline correction, by first or second derivatives; and normal signal standardization (Standard Normal Variate—SNV), to correct interferences related to the size of solid particles. Then, the data were mean centered. Reference values (polyphenol concentrations obtained by HPLC) were also mean centered. The selection of samples for the calibration and validation sets was performed using the Kennard–Stone algorithm. The number of latent variables (LV) was selected through leave-one-out cross-validation, based on a low value of the cross-validation error (Root Mean Square Error of Cross Validation—RMSECV). Root Mean Square Error of Calibration—RMSEC and validation errors (Root Mean Square Error of Prediction—RMSEP) were used to evaluate the performance of the models.

## 3. Results and Discussion

### 3.1. Evaluation of Polyphenols by Chromatography

Cyanidin-3-*O*-glucoside (C3G) and ellagic acid were the main polyphenols found in jabuticaba peel flour (Figure 2A,B), in agreement with previously reported studies [16,17]. Delphinidin-3-*O*-glucoside (D3G) was also identified (Figure 2A). These same compounds were also found in pectin extracted from jabuticaba peel flour [20]. Other compounds were separated during the chromatographic run, but could not be identified due to the absence of standard compounds. However, the separation and/or identification of gallic acid, epicatechin, ferulic acid and quercetin, which are expected to be found in jabuticaba peels based on the literature data [3,4], was not obtained in this study, despite the use of the standards to check for their presence. The non-detected polyphenols were gallic acid, epicatechin, ferulic acid and quercetin, which are low molecular weight polyphenols (as opposed to relatively higher molecular weight flavanols). These compounds are sparingly soluble in water and their solubility increases with temperature and with increasing mole fractions of alcohols, such as methanol and ethanol [21]. Given that the extraction method herein adopted employed an acidified 50% solution of ethanol in water, assisted by ultrasonication, the non-detection of these polyphenols by the HPLC method employed can only be justified by either their complete absence or their scarce presence in the studied matrix. Furthermore, flour preparation was carried out at 60 °C for 20 h. Studies on the thermal stability of gallic acid and catechin have demonstrated degradations from 15 to 20% to occur after 4 h of thermal treatment at 60 °C [22]. Thus, one should expect that the contents of the referred phenolics should be reduced in the flour in comparison with those of the fresh peels, and, because the thermal treatment at 60 °C in our work lasted 20 h, the contents might have been reduced to a point to which they would no longer be detected in the extract.

The C3G content ranged from 352.33 mg/100 g (S10) to 1008.73 mg/100 g JPF (S22) (Figure 3), representing a variation of 186.30%. These values were strongly correlated (r = 0.97; *p* = 0.00) with the content of total monomeric anthocyanins (TMA) (from 344.00 to 1304.52 mg/100 g), evaluated by the pH differential method and UV-vis in a previous study using the same samples [5]. Since C3G is the main anthocyanin present in jabuticaba peels, this correlation was expected; however, the C3G content showed a weak to moderate correlation (r = 0.45; *p* = 0.02) with the total extractable phenolics content (from 8.90 to 16.49 g gallic acid equivalents/100 g), based on the Folin–Ciocalteu (FC) method [5]. Although FC is not a specific method for phenolic compounds, since C3G is the main polyphenol present in jabuticaba peels, the low correlation suggests that the FC method is not very selective for C3G and may underestimate the antioxidant capacity of jabuticaba peels.

The C3G contents measured in the JPFs were lower than those reported by Quatrin et al. [3] in jabuticaba peel powders of the species *P. jaboticaba* (1039.1 mg/100 g) and *P. trunciflora* (1632 mg/100 g), and lower than that described by Plaza et al. [17] for *P. jaboticaba* peels (2866 mg/100 g of sample in dry weight), but higher than those found by Alezandro et al. [23] in *P. jaboticaba* (123 mg/100 g sample in dry weight). Barros et al. [16] reported that solutions acidified with formic acid are more effective for the recovery of anthocyanins in jabuticaba peels than those with acetic and phosphoric acid. Quatrin et al. [3] and Plaza et al. [17] used alcoholic solutions acidified with formic acid, unlike the present research and the study by Alezandro et al. [23], in which acetic acid was adopted.

Several health benefits of C3G have been observed in previous studies. Qi et al. [24] reported that black rice C3G supplementation prevented renal pathological changes in rats with diabetic nephropathy, inhibiting renal fibrosis and glomerular sclerosis by reducing blood glucose, improving insulin resistance and inhibiting renal oxidative stress and inflammatory cytokines. Hepatoprotective effects were reported by the use of C3G on liver damage induced by oxidative stress in mice [25]. Zhao et al. [26] observed that C3G extracted from blue honeysuckle was able to attenuate silica-induced pulmonary inflammatory responses in mice. Therefore, since JPFs are good sources of C3G, they can be viewed as food supplements with potential antioxidant and anti-inflammatory effects.

Delphinidin-3-*O*-glucoside was identified in the JPFs (Figure 2A), with contents ranging from 94.99 mg/100 g (S10) to 203.36 mg/100 g of JPF (S5) (Figure 4), showing a variation of 114.08%. However, no statistical difference was observed between most samples. Only samples S5, S24 and S21 were statistically different from S10. D3G levels showed a very strong correlation with total monomeric anthocyanins (r = 0.91; *p* = 0.00) and with C3G (r = 0.92; *p* = 0.00).

The average D3G content in the JPFs was similar to that reported in the study by Quatrin et al. [3] for jabuticaba peel powder (108.1 and 176 mg/100 g), and lower than that determined by Plaza et al. [17] (356.3 mg/100 g). The JPFs showed higher D3G content than mulberry samples of different cultivars (2.36 to 74.22 (mg/kg DW) [27]. D3G was described by Viegas et al. [28] as a promising nutritional agent against breast cancer, preventing tumor progression by inhibiting angiogenesis and showing selectivity towards cancer cells, which is a desirable trait.

Anthocyanins are well-known natural dyes [29]. Their contents obtained by HPLC correlated negatively and moderately with the color parameters of luminosity (L*) (ranging from 19.78 to 40.23) and hue angle (h*) (ranging from 8.58 to 21.47) reported for the same samples in a previous study [5]: C3G and L* (r = −0.65; *p* = 0.00); C3G and h* (r = −0.68; *p* = 0.00); D3G and L* (r = −0.64; *p* = 0.00); D3G and h* (r = −0.69; *p* = 0.00). These correlations were expected, since total monomeric anthocyanin content also correlated moderately with luminosity and hue angle, as reported by Resende et al. [5]. Therefore, peels that are darker and with a purple-red hue are expected to present higher amounts of C3G and D3G. This is in line with the characteristic color of cyanidins (magenta) and delphinidins (purple) [27]. The found correlations indicate that C3G and D3G have a significant influence on the darkness/lightness (luminosity) and tone (hue) of JPF, although they are not the only compounds to exert this influence. Anthocyanins did not show a correlation with the chromaticity data (c*) (ranging from 16.19 to 26.40), C3G (r = −0.05; *p* = 0.81) or D3G (r = −0.08; *p* = 0.68), which indicates that they do not interfere with the color saturation level.

The ellagic acid content ranged from 163.65 mg/100 g (S23) to 334.69 mg/100 g of JPF (S11) (Figure 5), showing a 104.52% variation. However, as observed for D3G, the ellagic acid contents of most samples were not statistically different. Samples S17 and S11 showed statistical difference from S23, which was also statistically different from S18 and S6.

A moderate to strong correlation (r = 0.69; *p* = 0.00) was observed between these data and the previously reported FC-based TEP results [5]. In this work, ellagic acid was the non-anthocyanin phenolic compound present in higher contents in JPF extracts. Considering that the FC method estimates the content of all phenolic compounds, the 69% correlation between these results is significant and demonstrates the importance of ellagic acid for the antioxidant capacity of JPF. It is important to highlight that the FC method, although widely used for providing a good estimate of the total phenolic content of most vegetables, is not specifically used for phenolic compounds. The reactivity of its reagent has been already shown for many other reducing compounds, including vitamins, unsaturated fatty acids, protein, carbohydrates, among others [30]. This may influence the correlation of extractable total phenolic data with the polyphenol contents evaluated by HPLC.

The levels of ellagic acid found in the twenty-eight JPF samples were higher than those reported for jabuticaba peels by Alezandro et al. [23] (40 mg/100 g), Quatrin et al. [3] (53.9 and 59.4 mg/100 g) and Plaza et al. [17] (142.8 mg/100 g). Acetic acid at pH 3 promoted a better recovery of phenolic compounds of jabuticaba peels than formic acid [16], used by Quatrin et al. [3] and Plaza et al. [17]. Ellagic acid contents were also higher than those determined for defatted raspberry seed extracts (43.05 to 46.76 mg/100 g) obtained by ultrasound-assisted extraction [31].

Aside from its antioxidant activity, ellagic acid is also associated with important health effects, such as those used in the treatment of chronic inflammatory diseases and conditions [32]. As described for C3G, ellagic acid also demonstrated protection against diabetic nephropathy in rats, with the prevention of oxidative stress, renal damage and inflammation in animals with induced diabetes mellitus [33]. Ellagic acid has also been shown to prevent arrhythmias and regulate the altered lipid profile in induced myocardial infarction in rats [34]. In a clinical study, ellagic acid and *Annona muricata* induced high-risk human papilloma virus (HR-HPV) clearance in women affected by low squamous intraepithelial lesion (L-SIL) related to HR-HPV [35]. Therefore, considering that JPFs are also good sources of ellagic acid, their regular consumption has the potential to contribute to the prevention and alleviation of symptoms of different diseases.

The distribution of the prepared flour samples, according to the polyphenol profile and the parameters reported in a previous study [5] that showed a significant correlation to the specific compounds as previously discussed, is presented in the biplot graph of the principal component analysis, PC1 vs. PC2, in Figure 6. Together, the two principal components explain 83.13% of data variability. PC1 explains 60.95% and separates the JPF with a higher polyphenol content from that with a lower polyphenol content, and that are also lighter and less reddish. PC2 explains 22.18% of data variability and separates the samples with more anthocyanins from those with higher amounts of ellagic acid and higher TEP values. The relationships between C3G, D3G, total monomeric anthocyanins, luminosity and hue, as well as ellagic acid and total extractable phenolics, can also be noted on the PCA graphic.

It is noteworthy that, although the extraction methods for HPLC, TMA and TEP were different, we are comparing distinct types of methodologies, and the extraction methods were selected according to the literature data for each specific type of analysis. While TEP and TMA provide an estimate of the total amounts of phenolics and anthocyanins, respectively, several studies have pointed out the limitations of such methods. HPLC, on the other hand, allows for the quantification of specific substances in a more reliable manner. The differences in extraction methods employed for HLPC, TMA and TEP have most certainly affected polyphenol yield and composition, so a direct comparison of the methods is not possible [36,37]. However, statistical analysis using Pearson’s correlation is still applicable, given that changes in the type of measurement should affect the agreement between techniques, but not the correlation [38,39]. The fact that high correlations were obtained between distinct methods, for example, in the case of C3G (the anthocyanin quantified in larger amounts) and TMA, indicates that TMA results can provide a reliable estimate of the anthocyanins in jabuticaba peels, regardless of the differences in extraction methodologies. The fact that C3G correlated poorly with TEP results indicates that other phenolics and, in addition, interfering substances are at play in this case, which may or may not be a result of using distinct extraction methodologies.

### 3.2. FTIR Analysis and Chemometric Models (PLS)

In Figure 7, the spectra of the 28 JPF samples can be seen in the fingerprint region (1800 to 600 cm^−1^), which is unique for each sample. Two bands associated to phenolics can be identified: 1390–1330 cm^−1^ and 1260–1180 cm^−1^, which result from the interaction between O–H angular deformation and C–O stretching [40]. For this reason, this region is important for the identification and quantification of phenolic compounds and was used as the starting point for the construction of prediction models for phenolic compounds.

Three PLS models were constructed to predict C3G, D3G and ellagic acid contents. Twenty-one samples were selected for the calibration set and seven samples were selected for validation. The parameters referring to the spectral range and pretreatments varied according to the best fit of the model. For the anthocyanins model, the spectral range of 1600–900 cm^−1^ was used and, for ellagic acid, 1800–600 cm^−1^. The FTIR data were subjected to the following pretreatments: first (D3G) and second derivatives (C3G and ellagic acid), and, for all models, SNV and mean centering. The PLS calibration parameters of the models are displayed in Table 2. All PLS models presented RMSEP/RMSEC ratios below three, indicating no overfitting of the models.

The best model was the one constructed for C3G prediction. The values obtained by the PLS model (predicted by FTIR data) versus values obtained by the HPLC method are presented in Figure 8A. The wavenumbers that contributed the most to the estimation of the C3G can be seen in the VIP scores graphic (Figure 8B) and include wavenumbers identified in the C3G spectrum [40], such as 1164 e 1016 cm^−1^, 1333 cm^−1^ and 1256 cm^−1^, which are in the characteristic range of phenols [41].

The comparison between predicted (FTIR) and measured (HPLC) contents of D3G and ellagic acid are presented in Figure 9A and Figure 10A, respectively. The wavenumbers that most influenced the D3G model (Figure 9B) are very similar to those of the C3G model (including 1346 cm^−1^), and include the band at 1043 cm^−1^, identified in the D3G spectrum by Choong, Mohd Yousof, Jamal and Isa Wasiman [41]. In the case of ellagic acid, the wavenumbers that mostly influenced the model (Figure 10B) include 1710 cm^−1^, previously identified in the ellagic acid spectrum by Tavares, Pena, Martin-Pastor and Sousa [42]. However, the wavenumbers that mostly influenced the model are in the range of 600–669 cm^−1^, a region in which the bands have no reported association with the functional groups in ellagic acid.

Even though the D3G and ellagic acid models have R² values lower than 0.9, they can still be considered satisfactory, since the differences between the predicted and measured values are small and may not be significant for routine analysis. It is noteworthy that chromatographic analysis demands much time and the consumption of toxic reagents, whereas FTIR spectroscopy is a non-destructive, fast and reliable technique that does not generate waste.

## 4. Conclusions

Three polyphenols with beneficial health activities were identified in flours obtained from jabuticaba peels, with significant differences among samples, as expected for native wild species. C3G was the polyphenol present in higher amounts, followed by ellagic acid and D3G for the majority of samples. The most pronounced differences among the samples in terms of phenolics were observed for C3G content. Strong correlations between C3G and D3G contents and total monomeric anthocyanins content were observed, along with moderate inverse correlations with respect to color parameters (luminosity and hue angle). Ellagic acid contents presented moderate to strong correlations with total extractable phenolic content measured by FC. PCA indicated the grouping of samples mainly according to polyphenol contents. Some differentiation was also observed between lighter samples with high ellagic acid amounts and darker samples with high anthocyanin contents. The PLS regression models based on FTIR spectroscopy data were able to provide satisfactory predictions of C3G, D3G and ellagic acid contents. 

## Figures and Tables

**Figure 1 foods-12-01488-f001:**
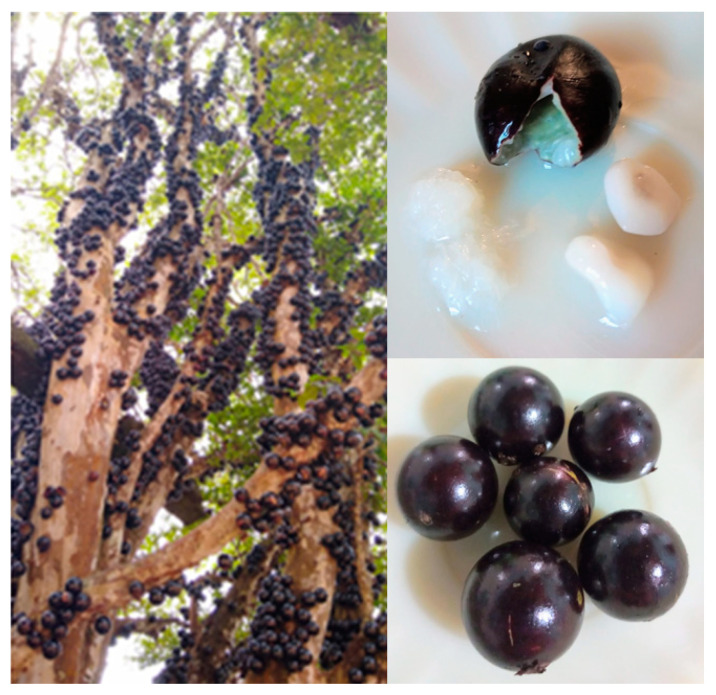
Photographs of jabuticaba fruits.

**Figure 2 foods-12-01488-f002:**
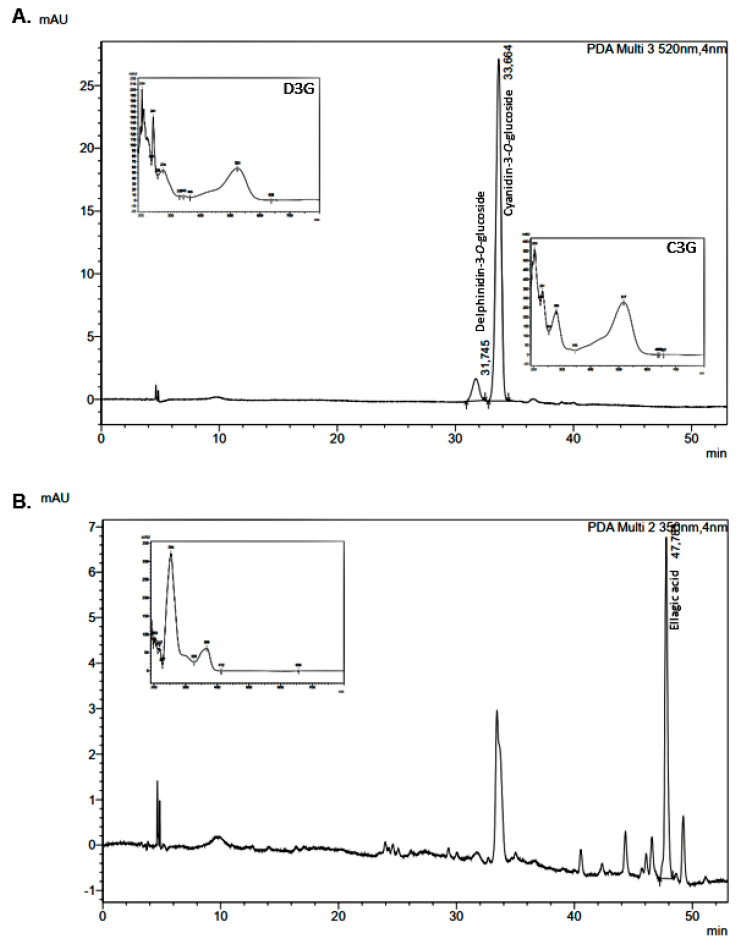
JPF chromatograms at (**A**) 520 and (**B**) 350 nm.

**Figure 3 foods-12-01488-f003:**
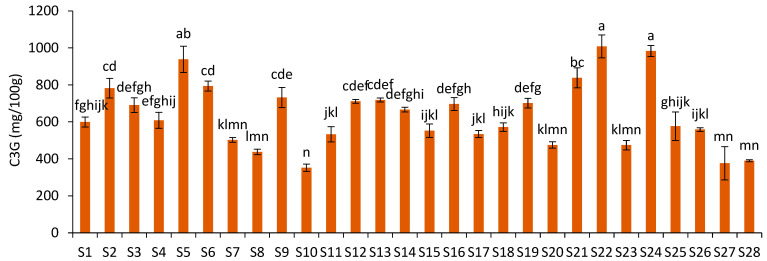
Cyanidin-3-*O*-glucoside content in different JPF samples. Different letters in different samples correspond to significantly different values (*p* < 0.05).

**Figure 4 foods-12-01488-f004:**
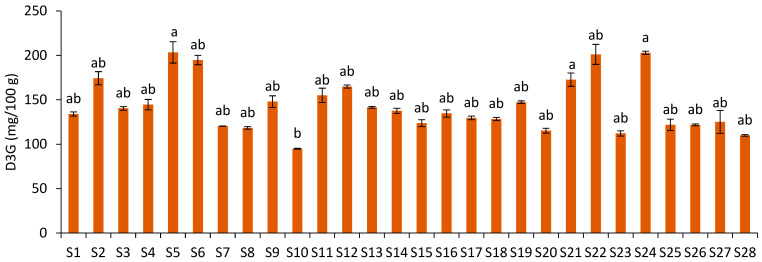
Delphinidin-3-*O*-glucoside content of JPF samples. Different letters in different samples correspond to significantly different values (*p* < 0.05).

**Figure 5 foods-12-01488-f005:**
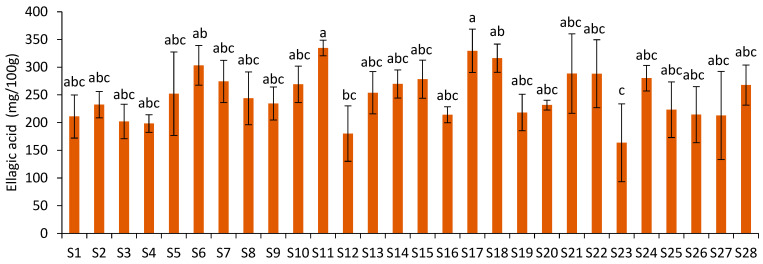
Ellagic acid content of JPF samples. Different letters in different samples indicate that values are significantly different (*p* < 0.05).

**Figure 6 foods-12-01488-f006:**
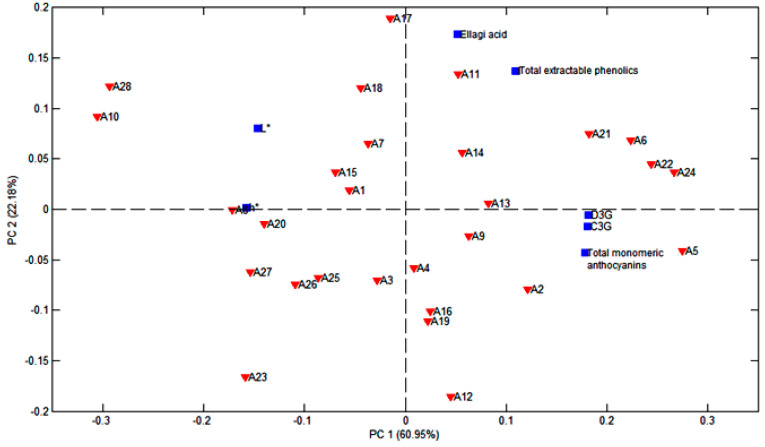
Biplot graphic PC1 vs. PC2 of polyphenol data and color parameters.

**Figure 7 foods-12-01488-f007:**
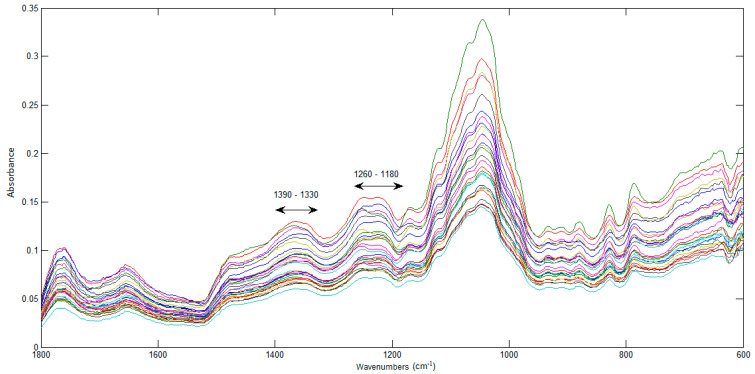
FTIR spectra of the jabuticaba peel flours.

**Figure 8 foods-12-01488-f008:**
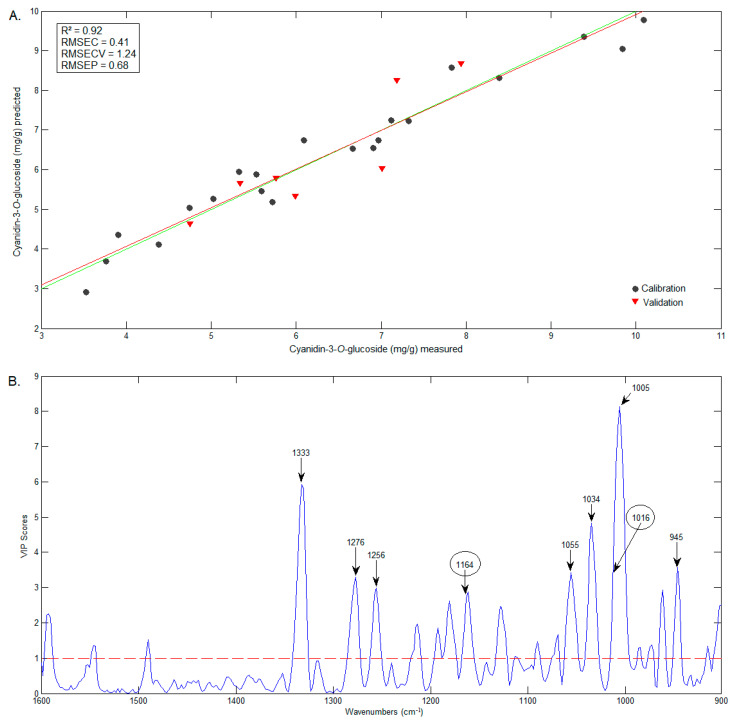
Graphics of the predicted vs. measured values (**A**) and of the VIP scores (**B**) of C3G.

**Figure 9 foods-12-01488-f009:**
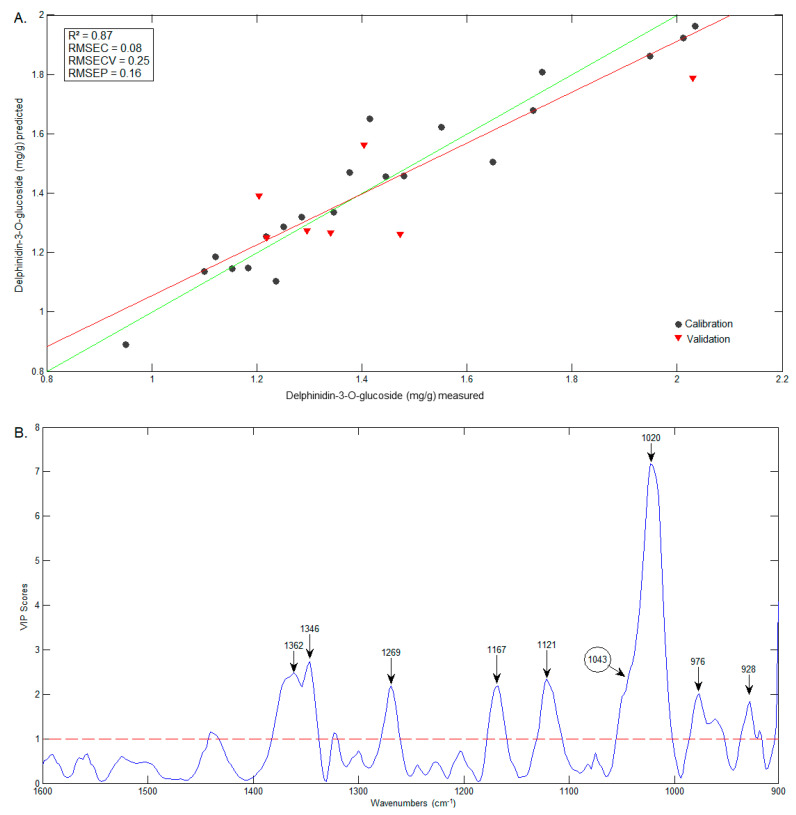
Graphics of the predicted vs. measured values (**A**) and of the VIP scores (**B**) of D3G.

**Figure 10 foods-12-01488-f010:**
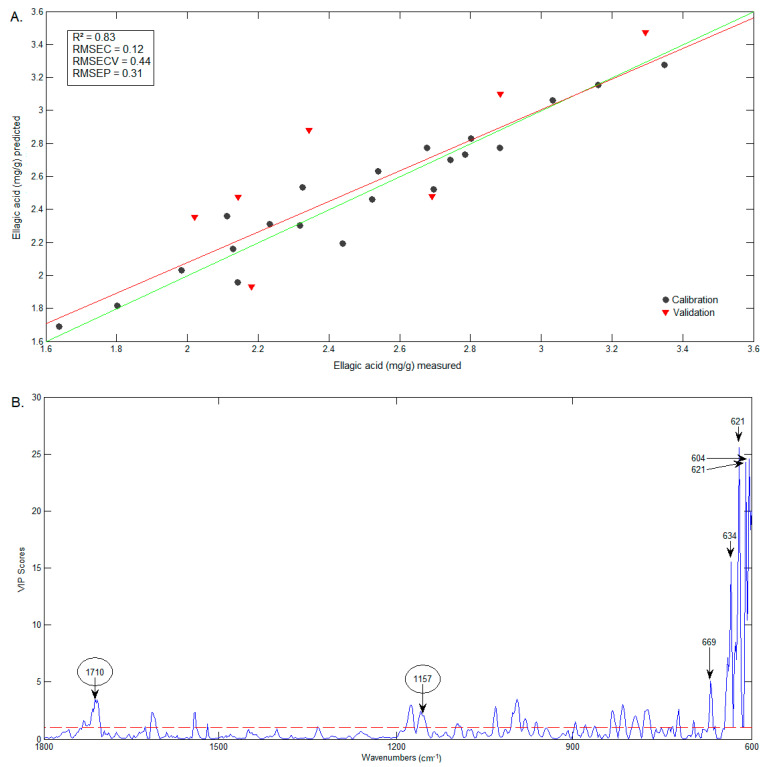
Graphics of the predicted vs. measured values (**A**) and of the VIP scores (**B**) of ellagic acid.

**Table 1 foods-12-01488-t001:** Elution gradient employed in the HPLC analysis.

Time (min)	% Solvent A	% Solvent B
0	95	5
5	95	5
45	50	50
53	50	50

**Table 2 foods-12-01488-t002:** Calibration parameters of models for predicting polyphenols in JPF.

Polyphenols	LV	R^2^	RMSEC	RMSECV	RMSEP
Cyanidin-3-*O*-glucoside	4	0.92	0.41	1.24	0.68
Delphinidin-3-*O*-glucoside	5	0.87	0.08	0.25	0.16
Ellagic acid	4	0.83	0.12	0.44	0.31

## Data Availability

The data presented in this study are available on request from the corresponding author.

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
