# Peer review of "Polyphenols in Jabuticaba (Plinia spp.) Peel Flours: Extraction and Comparative Evaluation of FTIR and HPLC for Quantification of Individual Compounds"

_foods, 2023, doi:10.3390/foods12071488_

Round 1

Reviewer 1 Report

1.      English language needs some revision and proofreading

Examples:  lines 45-46 Jabuticaba (Plinia sp.) (Figure 1), a Brazilian fruit, is an example dark colored fruits that are a source of polyphenols

Lines 64-65: but how much 64 may vary between samples currently available for consumption

2.      In abstract and in text in lines 251 and 181 correlations of HPLC results with TEP and TMA are mentioned as important ones, but methods fot TEP and TMA are not described anywere in this paper. Instead, the previously published paper is (incorrectly) cited.

3.      TEP and TMA values that are stated in lines 251 and 181 are the same as the values stated in the previous work of the same authors and it is obvious that this is the prolongation of the previous study on the same samples (https://doi.org/10.1016/j.lwt.2020.110135).

In their cited previous work the method used for phenolics extraction was with methanol and acetone, and in this paper the extraction method was with ethanol and acetic acid.

Since these 2 methods differ, it is not advisable to make correlations between the HPLC results obtained on one type of extracts with the Folin-Ciocalteau TEP values obtained on other type of extract, since the extraction conditions have significant influence on the phenolics content and profile (the authors themselves state the importance of the extraction method on the resuls in lines 66-67). This fact may compromise the correlations made in the text/abstract as well as biplot graphis in Figure 6.

Author Response

RESPONSE TO REVIEWER´S COMMENTS (in red)

English language needs some revision and proofreading

Examples:  lines 45-46 Jabuticaba (Plinia sp.) (Figure 1), a Brazilian fruit, is an example dark colored fruits that are a source of polyphenols

Lines 64-65: but how much 64 may vary between samples currently available for consumption

The texts in the examples were modified as requested. Furthermore, the manuscript was revised  by an English language teacher and modified accordingly.

  1. In abstract and in text in lines 251 and 181 correlations of HPLC results with TEP and TMA are mentioned as important ones, but methods fot TEP and TMA are not described anywere in this paper. Instead, the previously published paper is (incorrectly) cited.

Reference to the previous study was added to the abstract. TEP and TMA values were obtained from a previous study employing the same set of samples.

  1. TEP and TMA values that are stated in lines 251 and 181 are the same as the values stated in the previous work of the same authors and it is obvious that this is the prolongation of the previous study on the same samples (https://doi.org/10.1016/j.lwt.2020.110135).

The reviewer is correct, we believe that this is now clear in the study.

In their cited previous work the method used for phenolics extraction was with methanol and acetone, and in this paper the extraction method was with ethanol and acetic acid.

Since these 2 methods differ, it is not advisable to make correlations between the HPLC results obtained on one type of extracts with the Folin-Ciocalteau TEP values obtained on other type of extract, since the extraction conditions have significant influence on the phenolics content and profile (the authors themselves state the importance of the extraction method on the resuls in lines 66-67). This fact may compromise the correlations made in the text/abstract as well as biplot graphis in Figure 6.

Although the extraction methods for HLPC, TMA and TEP were different, we are comparing distinct types of methodology, and the extraction methods were selected according to the literature data for each specific type of analysis. While TEP and TMA provide an estimate of the total amounts of phenolics and anthocyanins, respectively, several studies have pointed out the limitations of such methods. HPLC, on the other hand, allows for the quantification of specific substances in a more reliable manner. Naturally, regardless of the differences in extraction methodologies, a direct comparison of the results is not possible. Nonetheless, the fact that high correlations were obtained between the two distinct methods, for example, in the case of C3G (the anthocyanin quantified in larger amounts) and TMA, indicates that TMA results can provide a reliable estimate of the anthocyanins in jabuticaba peels, regardless of the differences in extraction methodologies. The fact that C3G correlated poorly with TEP results indicates that other phenolics and also interfering substances are at play in this case, which may or may not be a result of using distinct extraction methodologies.

This information was added to the end of section 3.1.

The same reasoning applies for verifying correlations between HPLC results and color parameters, determinations are completely different (no need for extraction in the case of color measurements) but the correlations analysis is still applicable.

Reviewer 2 Report

Polyphenols in jabuticaba (Plinia spp.) peel: extraction and comparative evaluation of FTIR and HPLC for quantification of individual compounds

The article deals with an interesting topic. The article has the right structure. The experiments were planned correctly.

Notes below:

The abstract is too long. They should be shortened.

Why was the drying method used? Temperature affects polyphenols.

What was the purpose of the double centrifuge?

What exactly did the gradient look like in HPLC? The description is unclear. Please put it in the table.

Please give the equations of the standard curves in the HPLC method.

It is incomprehensible to compare the results to the FC method from another publication, in addition in terms of gallic acid, which is not included here at all. Please explain.

Why are only these specific polyphenols labeled? On what basis were the wavelengths at which the data collected selected? Why were other flavonoids and phenolic acids not determined?

Conclusions could relate to the FC method if the authors would use it in their work. On what grounds should these results converge or not? This method as a comparative method should be done in the work.

Author Response

RESPONSE TO REVIEWER´S COMMENTS (in red)

Polyphenols in jabuticaba (Plinia spp.) peel: extraction and comparative evaluation of FTIR and HPLC for quantification of individual compounds

The article deals with an interesting topic. The article has the right structure. The experiments were planned correctly.

Thank you for your comments.

Notes below:

The abstract is too long. They should be shortened.

The abstract word count was reduced from 387 to 303 words (approximately 20% size reduction)

Why was the drying method used? Temperature affects polyphenols.

Drying for preparation of flours from jabuticaba peels was used as a strategy in terms of conservation and application, given that it should reduce the free water content, increasing shelf life and safety of the product as a food ingredient, as well as concentrating the content of bioactive compounds. Although high temperatures can affect the polyphenol content, the conditions employed in our study were based on literature data for red grape pomace   (Larrauri, Rupérez & Saura-Calixto, 1997). This study concluded that drying at 60 °C would not affect significantly the total polyphenol content. The information about the purpose of drying was added to the end of section 2.2 in the revised manuscript.

Larrauri, J. A.; Rupérez, P.; Saura-Calixto, F. Effect of Drying Temperature on the Stability of Polyphenols and Antioxidant Activity of Red Grape Pomace Peels. J. Agric. Food Chem. 1997, 45, 1390–1393. https://doi.org/10.1021/jf960282f

What was the purpose of the double centrifuge?

We have verified that, after centrifugation, a residual amount of extracted polyphenols still remained adsorbed to the decanted solid particles. Therefore, to maximize the yield, further cycles of centrifugation were carried out and it was concluded that two centrifugation cycles were sufficient to separate the extracted polyphenols from their original solid matrix.

What exactly did the gradient look like in HPLC? The description is unclear. Please put it in the table.

The information was placed in a Table as requested.

Please give the equations of the standard curves in the HPLC method.

Added to the manuscript as requested.

It is incomprehensible to compare the results to the FC method from another publication, in addition in terms of gallic acid, which is not included here at all. Please explain.

TEP results are from a previous publication (Resende et al. 2020) using the same set of samples. Gallic acid is usually employed as a standard for building the FC calibration curve (Prior and Schaich, 2005), thus the results are presented in gallic acid equivalents. Since FC is commonly employed as a reference for total phenolics, the goal was to verify if this determination would have a direct correlation with the individual ammounts of phenolics.

Resende, L. M.; Oliveira, L. S.; Franca, A. S. Characterization of Jabuticaba (Plinia cauliflora) Peel Flours and Prediction of Compounds by FTIR Analysis. LWT 2020, 133 (August), 110135. https://doi.org/10.1016/j.lwt.2020.110135.

Prior, R. L.; Wu, X.; Schaich, K. Standardized Methods for the Determination of Antioxidant Capacity and Phenolics in Foods and Dietary Supplements. J. Agric. Food Chem. 2005, 53, 10, 4290–4302. https://doi.org/10.1021/jf0502698

Why are only these specific polyphenols labeled? On what basis were the wavelengths at which the data collected selected? Why were other flavonoids and phenolic acids not determined?

Previous literature studies (Inada et al., 2020; Rodrigues et al., 2015) have reported C3G, EA and D3G as the main phenolics found in jabuticaba peels, so we focused on these compounds. Other small peaks were detected but did not allow for quantification. Other substances that have been reported for jabuticaba peels include gallic acid, epicathechin, ferulic acid and quercetin (de Andrade Neves et al., 2021; Quatrin et al., 2019). We tested the standards but could not detect the specific peaks in our samples. This information is now presented in section 3.1.

The selected wavelengths were based the study by Plaza et al. (2016). Other wavelengths were tested, based on the publications of Quatrin et al. (2019), Rodrigues et al. (2015), but without significant results. This information was added to section 2.4, and is also referenced in section 3.1.

RODRIGUES, Sueli; FERNANDES, Fabiano A. N.; DE BRITO, Edy Sousa; SOUSA, Adriana Dutra; NARAIN, Narendra. Ultrasound extraction of phenolics and anthocyanins from jabuticaba peel. Industrial Crops and Products[S. l.], v. 69, p. 400–407, 2015. DOI: 10.1016/j.indcrop.2015.02.059. Disponível em: http://dx.doi.org/10.1016/j.indcrop.2015.02.059.

INADA, Kim Ohanna Pimenta; SILVA, Tamirys Barcellos Revorêdo; LOBO, Leandro Araújo; DOMINGUES, Regina Maria Cavalcante Pilotto; PERRONE, Daniel; MONTEIRO, Mariana. Bioaccessibility of phenolic compounds of jaboticaba (Plinia jaboticaba) peel and seed after simulated gastrointestinal digestion and gut microbiota fermentation. Journal of Functional Foods[S. l.], v. 67, n. November 2019, p. 103851, 2020. DOI: 10.1016/j.jff.2020.103851. Disponível em: https://doi.org/10.1016/j.jff.2020.103851

NEVES, Nathália de A.; STRINGHETA, Paulo C.; SILVA, Isadora, F. da; GARCÍA-ROMERO, Esteban; GÓMEZ-ALONSO, Sergio; HERMOSÍN-GUTIÉRREZ, Isidro. Identification and quantification of phenolic composition from different species of Jabuticaba (Plinia spp.) by HPLC-DAD-ESI/MSn. Food Chemistry[S. l.], v. 355, p. 129605, 2021. DOI: 10.1016/J.FOODCHEM.2021.129605.

QUATRIN, A. et al. Characterization and quantification of tannins, flavonols, anthocyanins and matrix-bound polyphenols from jaboticaba fruit peel: A comparison between Myrciaria trunciflora and M. jaboticaba. Journal of Food Composition and Analysis[S. l.], v. 78, n. January, p. 59–74, 2019. DOI: 10.1016/j.jfca.2019.01.018. Disponível em: https://doi.org/10.1016/j.jfca.2019.01.018.

Plaza, M.; Batista, Â. G.; Cazarin, C. B. B.; Sandahl, M.; Turner, C.; Östman, E.; Maróstica Júnior, M. R. Characterization of Antioxidant Polyphenols from Myrciaria Jaboticaba Peel and Their Effects on Glucose Metabolism and Antioxidant Status: A Pilot Clinical Study. Food Chem. 2016, 211, 185–197. https://doi.org/10.1016/j.foodchem.2016.04.142.

Conclusions could relate to the FC method if the authors would use it in their work. On what grounds should these results converge or not? This method as a comparative method should be done in the work.

As previously stated, all tests were performed using the same samples, but TEP and TMA results were already published.

Reviewer 3 Report

In this submitted paper, the authors tried to investigate specific polyphenols composition in flours obtained from jabuticaba peel (JPF) and examine the variation of such compounds content in samples prepared from some batches of fruits available for consumption. They also employed FTIR spectroscopy in combination with the chemometrics method for predicting the concentrations of polyphenols determined in the JPF. The subject of the paper is interesting, it's well-written and the discussion provided regarding the results obtained is acceptable. However, some minor issues underlined in the following comments need to be resolved before further consideration.

Minor issues:

1.      P.3, L.97, the extra '' grade'' has to be removed.

2. P.8, L.252, it would be a good idea to use the abbreviation ''FC'' here and anywhere on paper instead of the full term ''Folin-Ciocalteu''.

Author Response

RESPONSE TO REVIEWER´S COMMENTS (in red)

In this submitted paper, the authors tried to investigate specific polyphenols composition in flours obtained from jabuticaba peel (JPF) and examine the variation of such compounds content in samples prepared from some batches of fruits available for consumption. They also employed FTIR spectroscopy in combination with the chemometrics method for predicting the concentrations of polyphenols determined in the JPF. The subject of the paper is interesting, it's well-written and the discussion provided regarding the results obtained is acceptable. However, some minor issues underlined in the following comments need to be resolved before further consideration.

 Thank you for your comments.

Minor issues:

  1. P.3, L.97, the extra '' grade'' has to be removed.

Corrected as requested

  1. P.8, L.252, it would be a good idea to use the abbreviation ''FC'' here and anywhere on paper instead of the full term ''Folin-Ciocalteu''.

Corrected as requested

Reviewer 4 Report

Comments to the Author
The present study aimed to evaluate the composition of specific polyphenols in flours prepared from jabuticaba peel and employ FTIR together with chemometrics to predict the concentrations of these polyphenols in JPF.

However, I think the innovation of this study is seriously insufficient, because many studies use HPLC and FTIR methods to analyze the polyphenols in fruits and vegetables and their by -products. Such as Sivam, A. S., Sun-Waterhouse, D., Perera, C. O., & Waterhouse, G. I. N. (2012). Exploring the interactions between blackcurrant polyphenols, pectin and wheat biopolymers in model breads; a FTIR and HPLC investigation. Food Chemistry, 131(3), 802-810. Chupin, L., Motillon, C., Charrier-El Bouhtoury, F., Pizzi, A., & Charrier, B. (2013). Characterisation of maritime pine (Pinus pinaster) bark tannins extracted under different conditions by spectroscopic methods, FTIR and HPLC. Industrial Crops and Products, 49, 897-903. Awan, A. F., Akhtar, M. S., Anjum, I., Mushtaq, M. N., Fatima, A., Mannan, A., & Ali, I. (2020). Anti-oxidant and hepatoprotective effects of Lactuca serriola and its phytochemical screening by HPLC and FTIR analysis. Pakistan Journal of Pharmaceutical Sciences, 33. Etc…

Or maybe the author think the materials jabuticaba peel used in this study is novel? But But it can also be found that a lot of similar studies have been published such as Tarone, A. G., Silva, E. K., Cazarin, C. B. B., & Junior, M. R. M. (2021). Inulin/fructooligosaccharides/pectin-based structured systems: Promising encapsulating matrices of polyphenols recovered from jabuticaba peel. Food Hydrocolloids, 111, 106387. Cabral, B. R. P., de Oliveira, P. M., Gelfuso, G. M., Quintão, T. D. S. C., Chaker, J. A., de Oliveira Karnikowski, M. G., & Gris, E. F. (2018). Improving stability of antioxidant compounds from Plinia cauliflora (jabuticaba) fruit peel extract by encapsulation in chitosan microparticles. Journal of Food Engineering, 238, 195-201. Fernandes, I. D. A. A., Maciel, G. M., Maroldi, W. V., Bortolini, D. G., Pedro, A. C., & Haminiuk, C. W. I. (2022). Bioactive compounds, health-promotion properties and technological applications of Jabuticaba: A literature overview. Measurement: Food, 100057. Inada, K. O. P., Nunes, S., Martinez-Blazquez, J. A., Tomás-Barberán, F. A., Perrone, D., & Monteiro, M. (2020). Effect of high hydrostatic pressure and drying methods on phenolic compounds profile of jabuticaba (Myrciaria jaboticaba) peel and seed. Food Chemistry, 309, 125794.

Some other issue

1.   The abstract part of this manuscript needs to be modified to make it concise

2.   The overall content of this article needs to be re -combined to clearly display the research content

3.   The quality of the picture is very poor, and the picture should be combined to clearly reflect the author’s research results and content

4.   There are many contents in the manuscript quoting the literature to discuss the functionality of some polyphenols. I think it is meaningless, and at least it is not related to the theme of this study.

Author Response

RESPONSE TO REVIEWER´S COMMENTS (in red)

The present study aimed to evaluate the composition of specific polyphenols in flours prepared from jabuticaba peel and employ FTIR together with chemometrics to predict the concentrations of these polyphenols in JPF.

However, I think the innovation of this study is seriously insufficient, because many studies use HPLC and FTIR methods to analyze the polyphenols in fruits and vegetables and their by -products. Such as Sivam, A. S., Sun-Waterhouse, D., Perera, C. O., & Waterhouse, G. I. N. (2012). Exploring the interactions between blackcurrant polyphenols, pectin and wheat biopolymers in model breads; a FTIR and HPLC investigation. Food Chemistry, 131(3), 802-810. Chupin, L., Motillon, C., Charrier-El Bouhtoury, F., Pizzi, A., & Charrier, B. (2013). Characterisation of maritime pine (Pinus pinaster) bark tannins extracted under different conditions by spectroscopic methods, FTIR and HPLC. Industrial Crops and Products, 49, 897-903. Awan, A. F., Akhtar, M. S., Anjum, I., Mushtaq, M. N., Fatima, A., Mannan, A., & Ali, I. (2020). Anti-oxidant and hepatoprotective effects of Lactuca serriola and its phytochemical screening by HPLC and FTIR analysis. Pakistan Journal of Pharmaceutical Sciences, 33. Etc…

The reviewer is correct in the sense that there are many studies on the use of HPLC and FTIR for the analysis of polyphenols in fruits and vegetables. For instance, regarding the cited studies, the work by Sivam, A. S., Sun-Waterhouse, D., Perera, C. O., & Waterhouse, G. I. N. (2012) employed FTIR and HPLC for the chemical characterization of breads, and presented a very interesting discussion on the effect of variations in bread components on the FTIR spectra and composition of phenolics. Differences with respect to our study rely not only on the use of a completely different food matrix, but mainly on the fact that chemometrics was not employed by Sivam and co-workers, so no comparison of the techniques in terms of quantitative analysis was presented. The study by Chupin et al. (2013) focused on the characterization of pine bark and the comparison of different extraction techniques. While HPLC was employed for characterization of phenolics, FTIR was used for characterization regarding chemical functional groups and some generic classes of compounds, and the analyses focused on evaluating the effect of the different extraction procedures on the FTIR bands. No comparison of the techniques in terms of qualitative or quantitative analysis was performed. Finally, the study by Awan et al. (2020) focused on the phytochemical screening of lettuce (Lactuca serriola) by HPLC and FTIR. Again, FTIR was employed for characterization without a comparison of both techniques in terms of quantitative analysis.

Or maybe the author think the materials jabuticaba peel used in this study is novel? But But it can also be found that a lot of similar studies have been published such as Tarone, A. G., Silva, E. K., Cazarin, C. B. B., & Junior, M. R. M. (2021). Inulin/fructooligosaccharides/pectin-based structured systems: Promising encapsulating matrices of polyphenols recovered from jabuticaba peel. Food Hydrocolloids, 111, 106387. Cabral, B. R. P., de Oliveira, P. M., Gelfuso, G. M., Quintão, T. D. S. C., Chaker, J. A., de Oliveira Karnikowski, M. G., & Gris, E. F. (2018). Improving stability of antioxidant compounds from Plinia cauliflora (jabuticaba) fruit peel extract by encapsulation in chitosan microparticles. Journal of Food Engineering, 238, 195-201. Fernandes, I. D. A. A., Maciel, G. M., Maroldi, W. V., Bortolini, D. G., Pedro, A. C., & Haminiuk, C. W. I. (2022). Bioactive compounds, health-promotion properties and technological applications of Jabuticaba: A literature overview. Measurement: Food, 100057. Inada, K. O. P., Nunes, S., Martinez-Blazquez, J. A., Tomás-Barberán, F. A., Perrone, D., & Monteiro, M. (2020). Effect of high hydrostatic pressure and drying methods on phenolic compounds profile of jabuticaba (Myrciaria jaboticaba) peel and seed. Food Chemistry, 309, 125794.

Naturally we do not claim that jabuticaba peel is a novel material, since our own research group has already published other papers using this specific food by-product (see refs. 5 and 18 on the manuscript). Regarding the studies cited by the reviewer, we would like to point out several differences and highlight the contributions from our study. The paper by Tarone et al. (2021) focuses on the encapsulation of polyphenols from jabuticaba peel. There is no information regarding sample variability, so it is assumed that they were obtained from a specific batch (also, the deviation values on the reported averages of the measured parameters suggest that). We used samples from different municipalities and observed that the phenolic profiles presents a high variation, that might be associated to differences in micro-climatic and cultivation conditions as well as other environmental factors. Another difference is the quantification of phenolics by FTIR.  Cabral et al. (2018) also evaluated encapsulation of polyphenols from jabuticaba peel. Again, sample variability was not taken into account nor FTIR was performed. Similar differences are observed with respect to the study by Inada et al. (2020). The goal there was to evaluate the effect of different technological processes on the profile of phenolics in jabuticaba peel and seeds. The review by Fernandes et al. (2022) addresses the variability in fruit and peel composition related to different quantification techniques and extraction methods. Again, the variability associated to using a larger sample batch was not addressed. Furthermore, there is no reference to FTIR studies nor the application of chemometrics. 

In conclusion, it is our understanding that our paper presents innovative contributions, as previously pointed out. In summary, the major highlights/novelty of our work are:

  1. This is the first work that provides an investigation of specific polyphenols from a large sample of jabuticaba peels from the same region, thus reporting on the intrinsic variability among samples.
  2. This is the first work to correlate HPLC and FTIR measurements for quantification of specific compounds in jabuticaba peels, based on chemometric analysis and model building for a large sample of peels.

The introduction was modified in order to better reflect the contributions of the work.

Some other issue

  1. The abstract part of this manuscript needs to be modified to make it concise

The abstract word count was reduced from 387 to 303 words (approximately 20% size reduction)

  1. The overall content of this article needs to be re -combined to clearly display the research content

The manuscript was modified according to all the reviewers suggestions and recommendations.

  1. The quality of the picture is very poor, and the picture should be combined to clearly reflect the author’s research results and content

We do not have much control over the quality of the presented chromatograms, because they are automatically generated by the employed software, and there are also formating limitations because of the paper template. The files original figures were attached together with the manuscript, so improvements in figure quality can be attained during the editing phase.

  1. There are many contents in the manuscript quoting the literature to discuss the functionality of some polyphenols. I think it is meaningless, and at least it is not related to the theme of this study.

There are some references regarding the functional aspects of polyphenols and jabuticaba peel. Although this is not the goal of the study, it highlights the relevance and potential application of the studied food by-product.

Round 2

Reviewer 1 Report

1. Comments on the answers provided by the authors:

Although the extraction methods for HLPC, TMA and TEP were different, we are comparing distinct types of methodology, and the extraction methods were selected according to the literature data for each specific type of analysis. While TEP and TMA provide an estimate of the total amounts of phenolics and anthocyanins, respectively, several studies have pointed out the limitations of such methods. HPLC, on the other hand, allows for the quantification of specific substances in a more reliable manner. Naturally, regardless of the differences in extraction methodologies, a direct comparison of the results is not possible. Nonetheless, the fact that high correlations were obtained between the two distinct methods, for example, in the case of C3G (the anthocyanin quantified in larger amounts) and TMA, indicates that TMA results can provide a reliable estimate of the anthocyanins in jabuticaba peels, regardless of the differences in extraction methodologies. The fact that C3G correlated poorly with TEP results indicates that other phenolics and also interfering substances are at play in this case, which may or may not be a result of using distinct extraction methodologies.

This information was added to the end of section 3.1.

Reviewer comment:

The extraction solvent is one of the major determinants of polyphenol yield, polyphenol quantitative and qualitative composition. See recent papers from Carczorova et al (10.3390/molecules26061601), Baron G eta al (DOI: 10.3390/molecules26185454) for qualitative and antioxidant properties of extracts obtained with different solvents, and papers from Do QD eta al (https://doi.org/10.1016/j.jfda.2013.11.001) and Plaudo MC et al (https://doi.org/10.21577/0103-5053.20190047 ) for quantitative (TEP and TMA) yields depending on different solvents.

Before making the correlations that were made in the paper using different extraction solvents, the authors have to find the base in existing literature  that support this, or to make their own small study testing this hypothesis about the possibility of comparing the results of total polyphenols (TEP) and TMA obtained with one solvent with the HPLC results of the same polyphenols obtained with other solvent.

The same reasoning applies for verifying correlations between HPLC results and color parameters, determinations are completely different (no need for extraction in the case of color measurements) but the correlations analysis is still applicable.

Reviewer comment:

This is not acceptable explanation, because color in plant parts is the result of different colored components, not just polyphenols, so the correlation that is being made is of less strickt nature than correlations that are being made on polyphenols profiles/contents only.

Other comments:

Introduction

In Introduction whole corrected paragraph in lines 63-71 is not directly correlated to the other data concerning used samples in the paper - it is not clear if the samples came from different jabuticaba species or the same species but from distinct places within the region (lines 63-71) or different batches of the same species of fruit available for consumption (lines 90-91) which is stated as one of the aims of the investigation. This should be described with more detail and if the species are known it shoud be also stated in the paper.

This also relates to the section 2.1 and sentence „Twenty-eight jabuticaba samples were acquired from nineteen municipalities in Minas Gerais State, Brazil“ – this needs more information and clarification if the species and other origin characteristics are important for the pbjective of the study as stated in lines 90-91.

Materials and Methods

In Section 2.2 Sample preparation the drying process of jabuticaba peels is desribed. Why „in a convective oven at 60 °C for 20 h.“ is used? It seems like a long exposure to increased temperature that could influence the qualitative and quantitative composition of polyphenols. Is there an explanation for using these particular conditions?

Results and Discussion

3.1 The statement about non-identifying some expected polyphenols given in lines 194-197 needs some explanation. What could be the reason – different extraction methods, different species, lower sensitivity of used HPLC method/column...?

Author Response

RESPONSE TO REVIEWER´S COMMENTS (in red)

The extraction solvent is one of the major determinants of polyphenol yield, polyphenol quantitative and qualitative composition. See recent papers from Carczorova et al (10.3390/molecules26061601), Baron G eta al (DOI: 10.3390/molecules26185454) for qualitative and antioxidant properties of extracts obtained with different solvents, and papers from Do QD eta al (https://doi.org/10.1016/j.jfda.2013.11.001) and Plaudo MC et al (https://doi.org/10.21577/0103-5053.20190047 ) for quantitative (TEP and TMA) yields depending on different solvents.

Before making the correlations that were made in the paper using different extraction solvents, the authors have to find the base in existing literature  that support this, or to make their own small study testing this hypothesis about the possibility of comparing the results of total polyphenols (TEP) and TMA obtained with one solvent with the HPLC results of the same polyphenols obtained with other solvent.

This is not acceptable explanation, because color in plant parts is the result of different colored components, not just polyphenols, so the correlation that is being made is of less strickt nature than correlations that are being made on polyphenols profiles/contents only.

We understand the reviewer´s concern and believe that this should have been better explained in our manuscript. As mentioned by the reviewer, differences in extraction methods will affect polyphenol yield and composition, thus hindering the possibility of a direct comparison of the determinations. However, statistical analysis using Pearsons correlation is still applicable, given that, even though changes in the type of measurement should affect the agreement between techniques, correlation results remain unaltered (Blan and Altman, 1986). For instance, a recent study (Jara et al. 2022) evaluated correlations between TPC measurements and the contents of phenolics determined by LC-MS. Different extraction solution were used for each method and, based on correlation analyses, the authors were able to evaluate which specific compounds had a positive correlation with TPC values regardless of the extraction method. Another study (Ye et al. 2023) employed Pearson´s analysis to verify if there was any correlation between bioactive compounds and minerals in Se-enriched green tea. Two distinct techniques were used, namely ICP-MS and HPLC for determining the concentration of minerals and organic substances (alkaloids and polyphenols), respectively. They concluded that Se contents in tea leaves were positively associated with gallic acid and gallated catechins (including EGCG, GCG, and ECG) and inversely associated with non-gallated catechins, suggesting that Se might be a participator in regulating the catechin biosynthesis. Our goal here was to apply the same type of reasoning. Although extraction methods as well as measurement techniques (spectroscopic methods vs.  HLPC) were completely different, the correlation analysis is still feasible. Notice that distinct parameters are being compared, namely TPC/TMA and concentration of individual parameters. The same applies for verifying correlations between HPLC results and color parameters, determinations are completely different (no need for extraction in the case of color measurements) but the correlation analysis is still applicable.

In order to clarify this matter, we modified the last paragraph of section 7.1 and added the following corresponding references (the ones previously cited in our explanation and others recommended by the reviewer):

“It is noteworthy to mention that, although the extraction methods for HPLC, TMA and TEP were different, we are comparing distinct types of methodologies, and the extrac-tion methods were selected according to the literature data for each specific type of analy-sis. While TEP and TMA provide an estimate of the total amounts of phenolics and an-thocyanins, respectively, several studies have pointed out the limitations of such methods. HPLC, on the other hand, allows for the quantification of specific substances in a more reliable manner. Naturally, regardless of the differences in extraction methodologies, a direct comparison of the results is not possible. The differences in extraction methods employed for HLPC, TMA and TEP have most certainly affected polyphenol yield and composition, so a direct comparison of the methods is not possible [34, 35]. However, statisti-cal analysis using Pearsons correlation is still applicable, given that changes in the type of measurement should affect the agreement between techniques but not the correlation [36-38]. The fact that high correlations were obtained between distinct methods, for exam-ple, in the case of C3G (the anthocyanin quantified in larger amounts) and TMA, indicates that TMA results can provide a reliable estimate of the anthocyanins in jabuticaba peels, regardless of the differences in extraction methodologies. The fact that C3G correlated poorly with TEP results indicates that other phenolics and also interfering substances are at play in this case, which may or may not be a result of using distinct extraction methodologies.”

[32] Baron, G.; Ferrario, G.; Marinello, C.; Carini, M.; Morazzoni, P.; Aldini, G. Effect of Extraction Solvent and Temperature on Polyphenol Profiles, Antioxidant and Anti-Inflammatory Effects of Red Grape Skin By-Product. Molecules 2021, 26, 5454. https://doi.org/10.3390/molecules26185454

[33] Do, Q.D.; Angkawijaya, A.E.; Tran-Nguyen, P.L.; Huynh, L.H.; Soetaredjo, F.E.; Ismadji, S.; Ju, Y.-H. Effect of extraction solvent on total phenol content, total flavonoid content, and antioxidant activity of Limnophila aromatica, J. Food Drug Anal. 2014, 22, 296-302, https://doi.org/10.1016/j.jfda.2013.11.001.

[34] Bland, J.M.; Altman, D.G. Statistical methods for assessing agreement between two methods of clinical measurement, The Lancet 1986, 327, 307-310, https://doi.org/10.1016/S0140-6736(86)90837-8.

[35] Ye, Y.; Yan, W.; Peng, L.; He, J.;  Zhang, N.; Zhou, J.; Cheng, S.; Cai ,J. Minerals and bioactive components profiling in Se-enriched green tea and the Pearson correlation with Se, LWT 2023, 175, 114470, https://doi.org/10.1016/j.lwt.2023.114470.

[36] Jara, F.M.;  Carrión, M.E.; Angulo, J.L.; Latorre, G.; López-Córcoles, H.; Zalacain, A.; de Mendoza, J.H.; García-Martínez, M.M.; ;Carmona, M. Chemical characterization, antioxidant activity and morphological traits in the leaves of guayule (Parthenium argentatum A. Gray) and its hybrids, Ind. Crops Prod. 2022, 182, 114927, https://doi.org/10.1016/j.indcrop.2022.114927.

Other comments:

Introduction

In Introduction whole corrected paragraph in lines 63-71 is not directly correlated to the other data concerning used samples in the paper - it is not clear if the samples came from different jabuticaba species or the same species but from distinct places within the region (lines 63-71) or different batches of the same species of fruit available for consumption (lines 90-91) which is stated as one of the aims of the investigation. This should be described with more detail and if the species are known it shoud be also stated in the paper.

This also relates to the section 2.1 and sentence „Twenty-eight jabuticaba samples were acquired from nineteen municipalities in Minas Gerais State, Brazil“ – this needs more information and clarification if the species and other origin characteristics are important for the pbjective of the study as stated in lines 90-91.

The samples are from the same species and come from distinct micro-regions of Minas Gerais State, Brazil. Detailed information on sample origin was added as supplementary data and the text in lines 90-91 was modified to avoid confusion.

Materials and Methods

In Section 2.2 Sample preparation the drying process of jabuticaba peels is desribed. Why „in a convective oven at 60 °C for 20 h.“ is used? It seems like a long exposure to increased temperature that could influence the qualitative and quantitative composition of polyphenols. Is there an explanation for using these particular conditions?

The goal of this study was to use drying as a strategy in terms of conservation and application, in order to increase the shelf life and safety of the product as a food ingredient, using a conventional and cheap conservation approach (freeze-drying, for instance, would probably be safer in terms of affecting the composition). We added the term “flours” to the manuscript title in order to clarify that our food matrix is actually a flour and not the peel itself. Thus, the product of interest is the flour dried under the specified conditions. We also added a more recent publication regarding the production of fruit and vegetable by-products' flours that confirms 50-60oC and 4-48h as the most commonly employed temperature and time ranges during the drying step. The paragraph now reads:

“It is noteworthy to point out that drying was used as a strategy in terms of conservation and application, in order to increase the shelf life and safety of the product as a food ingredient, and also to concentrate the content of bioactive compounds. Although high temperatures can affect the polyphenol content, the conditions employed in our study were based on the findings reported by Larrauri et al. [14] on the effect of drying temperature on the stability of polyphenols in red grape pomace, and are within the range that is commonly employed for producing flours based on fruit and vegetable by-products [16].”

[14] Larrauri, J.A., Rupérez, P. and Saura-Calixto, F. Effect of drying temperature on the stability of polyphenols and antioxidant activity of red grape pomace peels. J Agric.  Food Chem. 1997, 45, 1390-1393. http://dx.doi.org/10.1021/jf960282f

[15] Santos, D.; da Silva, J.A. L.; Manuela Pintado, M. Fruit and vegetable by-products' flours as ingredients: a review on production process, health benefits and technological functionalities, LWT 2022, 154, 112707, https://doi.org/10.1016/j.lwt.2021.112707

Results and Discussion

3.1 The statement about non-identifying some expected polyphenols given in lines 194-197 needs some explanation. What could be the reason – different extraction methods, different species, lower sensitivity of used HPLC method/column...?

The non-detected polyphenols were gallic acid, epicatechin, ferulic acid and quercetin, which are low molecular weight polyphenols (as opposed to relatively higher molecular weight flavanols). These compounds are sparingly soluble in water and their solubility increases with temperature and with increasing mole fractions of alcohols, such as methanol and ethanol (Daneshfar et al., 2008; Razmara et al., 2010; Cuevas-Valenzuela et al., 2014). Given that the extraction method adopted in our work employed an acidified 50 % solution of ethanol in water, assisted by ultrasonication, and the fact that low molecular weight alcohols (such as ethanol and methanol) are best for extracting polyphenols with relatively lower molecular weights (Do et al., 2014), the non-detection of these polyphenols by the HPLC method employed can only be justified by either their complete absence or scarce presence in the studied matrix. Furthermore, the preparation of the flour was carried out at 60 oC for 20 h and studies on the thermal stability of gallic acid and catechin have demonstrated degradations of 15 to 20 % to occur after 4 h of thermal treatment at 60 oC (Volf et al., 2014). Thus, one should expect that the contents of the referred phenolics should be reduced in the flour in comparison with those of the fresh peels, and, because the thermal treatment at 60 oC in our work lasted 20 h, the respective contents might have been reduced to a point of not being detected in the extract. Recall that the standards for the referred phenolics were duly employed in the HPLC analysis.

Cuevas-Valenzuela, J., González-Rojas, A., Wisniak, J., Apelblat, A., Pérez-Correa, J.R. Solubility of (+)-catechin in water and water-ethanol mixtures within the temperature range 277.6–331.2K: Fundamental data to design polyphenol extraction processes. Fluid Phase Equilibria, Volume 382, 2014, 279-285. https://doi.org/10.1016/j.fluid.2014.09.013.

Daneshfar, A., Ghaziaskar, H.S., Homayoun, N. Solubility of Gallic Acid in Methanol, Ethanol, Water, and Ethyl Acetate. Journal of Chemical & Engineering Data 2008 53 (3), 776-778. https://doi.org/10.1021/je700633w

Do, Q., Angkawijaya, A., Tran-Nguyen, P., Huynh, L., Soetaredjo, F., Ismadji, S., Ju. Y.H. 2014. Effect of extraction solvent on total phenol content, total flavonoid content, and antioxidant activity of Limnophila aromatica. J Food Drug Anal 22(3): 296-302. https://doi.org/10.1016/j.jfda.2013.11.001

Razmara, R.S., Daneshfar, A., Sahraei, R. Solubility of Quercetin in Water + Methanol and Water + Ethanol from (292.8 to 333.8) K. Journal of Chemical & Engineering Data, 2010, 55 (9), 3934-3936. https://doi.org/10.1021/je9010757

Volf, I., Ignat, I., Neamtu, M., Popa, V.I. Thermal stability, antioxidant activity, and photo-oxidation of natural polyphenols. Chem. Pap. 68, 121–129 (2014). https://doi.org/10.2478/s11696-013-0417-6

In order to clarify this, the following text and respective references were added to the manuscript

“The non-detected polyphenols are low molecular weight polyphenols (as opposed to rela-tively higher molecular weight sparingly soluble in water, but their solubility increases with temperature and with increasing mole fractions of alcohols, such as methanol and ethanol [21]. Given that the extraction method herein adopted employed an acidified 50 % solution of ethanol in water, assisted by ultrasonication, the non-detection of these poly-phenols by the HPLC method employed can only be justified by either their complete ab-sence or scarce presence in the studied matrix. Furthermore, flour preparation was carried out at 60 oC for 20 h.  Studies on the thermal stability of gallic acid and catechin have demonstrated degradations of 15 to 20 % to occur after 4 h of thermal treatment at 60 oC [22]. Thus, one should expect that the contents of the referred phenolics should be reduced in the flour in comparison with those of the fresh peels, and, because the thermal treat-ment at 60 oC in our work lasted 20 h, the respective contents might have been reduced to a point of not being detected in the extract.”

Reviewer 2 Report

The Authors introduced several changes suggested by the reviewer. However, there are still some very important issues that have not been clarified. The explanation regarding the method of drying to obtain the preparation (freeze-drying method would certainly be better) is insufficient. As the lack of changes in the composition of polyphenols in the tested matrix, the Authors refer to the publication from 1997. Back then, there wasn't the analytics that there is now. Please find a better justification. I still think it's wrong to compare the markings of the two articles. If these markings cannot be repeated, then perhaps this article should be treated as the second part of that one? Expressing the total amount of polyphenols in terms of gallic acid is accepted in the literature in preparations or products in which this acid is present in large amounts.

Author Response

RESPONSE TO REVIEWER´S COMMENTS (in red)

The Authors introduced several changes suggested by the reviewer. However, there are still some very important issues that have not been clarified. The explanation regarding the method of drying to obtain the preparation (freeze-drying method would certainly be better) is insufficient. As the lack of changes in the composition of polyphenols in the tested matrix, the Authors refer to the publication from 1997. Back then, there wasn't the analytics that there is now. Please find a better justification.

Freeze-drying would definitely be a more appropriate method for reducing moisture content without damaging product composition. However, the goal here was to use drying as a strategy in terms of conservation and application, in order to increase the shelf life and safety of the product as a food ingredient, using a more conventional and cheaper conservation approach. We added the term “flours” to the manuscript title in order to clarify that our food matrix is actually a flour and not fresh or untreated peel itself. We also added a more recent publication regarding the production of fruit and vegetable by-products' flours that confirms 50-60 oC as the most commonly employed temperature ranges during the drying step.

Santos, D., Silva, J.A.L., Pintado, M. Fruit and vegetable by-products' flours as ingredients: A review on production process, health benefits and technological functionalities. LWT, Volume 154, 2022, 112707. https://doi.org/10.1016/j.lwt.2021.112707

The corrected paragraph in section 2.2 is as follows:

“It is noteworthy to point out that drying was used as a strategy in terms of conservation and application, in order to increase the shelf life and safety of the product as a food ingredient, and also to concentrate the content of bioactive compounds. Although high temperatures can affect the polyphenol content, the conditions employed in our study were based on the findings reported by Larrauri et al. [14] on the effect of drying temperature on the stability of polyphenols in red grape pomace, and are within the range that is commonly employed for producing flours based on fruit and vegetable by-products [15].”

I still think it's wrong to compare the markings of the two articles. If these markings cannot be repeated, then perhaps this article should be treated as the second part of that one? Expressing the total amount of polyphenols in terms of gallic acid is accepted in the literature in preparations or products in which this acid is present in large amounts.

We are comparing our previously published results on TEP and TMA with HPLC determination by means of Pearson´s correlation.  As pointed out by another reviewer, differences in extraction methods will affect polyphenol yield and composition, thus hindering the possibility of a direct comparison of the determinations. However, statistical analysis using Pearson´s correlation is still applicable, given that, even though changes in the type of measurement should affect the agreement between techniques, correlation results remain unaltered (Blan and Altman, 1986). For instance, a recent study (Jara et al. 2022) evaluated correlations between TPC measurements and the contents of phenolics determined by LC-MS. Different extraction solution were used for each method and, based on correlation analyses, the authors were able to evaluate which specific compounds had a positive correlation with TPC values regardless of the extraction method.

Both the present manuscript and the already published article are part of a concluded Ph.D. thesis, so in that sense we could view this manuscript as a continuation. Nonetheless, we believe that it would be strange to name this as part II, since the published article is from another journal. Furthermore, the names of both publications would have to be changed. TEP data were already published using gallic acid equivalents, so we didn´t think there was a need to modify the units, because the correlation analysis that was one of the goals of this study would not be modified.

The following discussion was added to the last paragraph of section 3.1:

“It is noteworthy to mention that, although the extraction methods for HPLC, TMA and TEP were different, we are comparing distinct types of methodologies, and the extrac-tion methods were selected according to the literature data for each specific type of analy-sis. While TEP and TMA provide an estimate of the total amounts of phenolics and an-thocyanins, respectively, several studies have pointed out the limitations of such methods. HPLC, on the other hand, allows for the quantification of specific substances in a more reliable manner. Naturally, regardless of the differences in extraction methodologies, a direct comparison of the results is not possible. The differences in extraction methods employed for HLPC, TMA and TEP have most certainly affected polyphenol yield and composition, so a direct comparison of the methods is not possible [34, 35]. However, statisti-cal analysis using Pearsons correlation is still applicable, given that changes in the type of measurement should affect the agreement between techniques but not the correlation [36-38]. The fact that high correlations were obtained between distinct methods, for exam-ple, in the case of C3G (the anthocyanin quantified in larger amounts) and TMA, indicates that TMA results can provide a reliable estimate of the anthocyanins in jabuticaba peels, regardless of the differences in extraction methodologies. The fact that C3G correlated poorly with TEP results indicates that other phenolics and also interfering substances are at play in this case, which may or may not be a result of using distinct extraction methodologies.”

[32] Baron, G.; Ferrario, G.; Marinello, C.; Carini, M.; Morazzoni, P.; Aldini, G. Effect of Extraction Solvent and Temperature on Polyphenol Profiles, Antioxidant and Anti-Inflammatory Effects of Red Grape Skin By-Product. Molecules 2021, 26, 5454. https://doi.org/10.3390/molecules26185454

[33] Do, Q.D.; Angkawijaya, A.E.; Tran-Nguyen, P.L.; Huynh, L.H.; Soetaredjo, F.E.; Ismadji, S.; Ju, Y.-H. Effect of extraction solvent on total phenol content, total flavonoid content, and antioxidant activity of Limnophila aromatica, J. Food Drug Anal. 2014, 22, 296-302, https://doi.org/10.1016/j.jfda.2013.11.001.

[34] Bland, J.M.; Altman, D.G. Statistical methods for assessing agreement between two methods of clinical measurement, The Lancet 1986, 327, 307-310, https://doi.org/10.1016/S0140-6736(86)90837-8.

[35] Ye, Y.; Yan, W.; Peng, L.; He, J.;  Zhang, N.; Zhou, J.; Cheng, S.; Cai ,J. Minerals and bioactive components profiling in Se-enriched green tea and the Pearson correlation with Se, LWT 2023, 175, 114470, https://doi.org/10.1016/j.lwt.2023.114470.

[36] Jara, F.M.;  Carrión, M.E.; Angulo, J.L.; Latorre, G.; López-Córcoles, H.; Zalacain, A.; de Mendoza, J.H.; García-Martínez, M.M.; ;Carmona, M. Chemical characterization, antioxidant activity and morphological traits in the leaves of guayule (Parthenium argentatum A. Gray) and its hybrids, Ind. Crops Prod. 2022, 182, 114927, https://doi.org/10.1016/j.indcrop.2022.114927.

Reviewer 4 Report

Even though the author has made some modifications and explanatory notes in the new manuscript, I believe that these modifications are limited and still fail to adequately demonstrate the originality of this research. Additionally, the quality of the figures and charts is poor and does not meet publication standards. Despite the fact that many of the charts were exported from instruments, the author could still optimize and improve them appropriately.

Author Response

RESPONSE TO REVIEWER´S COMMENTS (in red)

Even though the author has made some modifications and explanatory notes in the new manuscript, I believe that these modifications are limited and still fail to adequately demonstrate the originality of this research.

We added the following paragraph to the introduction in order to clarify the main contributions of our work:

In summary, the major contributions of this study are: (i) this is the first work that provides an investigation of specific polyphenols from a large sample of jabuticaba peels from distinct micro-regions of the same Brazilian State, thus reporting on the intrinsic variability of the samples attributed to differences in edaphoclimatic conditions and agri-cultural practices. (ii) This is the first work to correlate HPLC and FTIR measurements for quantification of specific compounds in jabuticaba peel flours, including chemometric analysis and model building for a large sample of peels.

The last three paragraphs of the introduction are focused on stating the relevance and originality of the study, so we hope that it is better now. We have also extensively modified the manuscript text and added new references  in order to better explain the methodology and results (see red markings at the left margin regarding all changes).

Additionally, the quality of the figures and charts is poor and does not meet publication standards. Despite the fact that many of the charts were exported from instruments, the author could still optimize and improve them appropriately.

Figures and charts were modified accordingly. Some were modified in the original files and some were processed in Adobe Photoshop in order to improve contrast. The original figures can also be sent later as separate files if necessary, so quality can be better assessed by the editorial office when processing the manuscript.